# From By-Products to Fertilizer: Chemical Characterization Using UPLC-QToF-MS via Suspect and Non-Target Screening Strategies

**DOI:** 10.3390/molecules27113498

**Published:** 2022-05-29

**Authors:** Anthi Panara, Evagelos Gikas, Nikolaos S. Thomaidis

**Affiliations:** Laboratory of Analytical Chemistry, Department of Chemistry, National and Kapodistrian University of Athens, 15771 Athens, Greece; panaranthi@chem.uoa.gr (A.P.); vgikas@chem.uoa.gr (E.G.)

**Keywords:** onion, mushroom, fertilizer, UPLC-QToF-MS, suspect screening, non-target screening

## Abstract

The increasing demands of agriculture and the food market have resulted in intensive agricultural practices using synthetic fertilizers to maximize production. However, significant efforts have been made to implement more environmentally friendly procedures, such as composting, to overcome the adverse impact of these invasive practices. In the terms of this research, composting was applied to the production of two biofertilizers, using onion and mushroom by-products as raw materials respectively. The main purposes of this work were to identify the compounds that pass from the raw materials to the final products (onion-based and mushroom-based), as well as the characterization of the chemical profile of these final products following suspect and non-target screening workflows via UPLC-qToF-MS. Overall, 14 common compounds were identified in the onion and its final product, while 12 compounds were found in the mushroom and its corresponding product. These compounds belong to fatty acids, organic acids, and flavonoids, which could be beneficial to plant health. The determination of parameters, such as the pH, conductivity, organic matter, nitrogen content, and elemental analysis, were conducted for the overall characterization of the aforementioned products.

## 1. Introduction

The demands of the agri-food market in contemporary society have resulted in the global and national proliferation of intensive agricultural practices. In this direction, the use of synthetic fertilizers has increased in order to maximize their production. However, chemical fertilizer utilization that exceeds a certain threshold level pollutes water bodies in addition to causing accumulation in crop plants [1]. Therefore, considerable effort has been expended to seek more environmentally friendly methods of increasing productivity while also ensuring agricultural biosafety with no adverse effects on soil microflora, the environment, or human health.

Composting is the aerobic decomposition of organic materials into a humus-like material through microbial action [2]. Research found that soil supplementation with compost acting as fertilizer is beneficial towards improving the physical chemical properties and nutrient status [3]. Furthermore, compost is a natural and environmentally friendly product that is well known for its beneficial effects on soil and plant nutrition, whereas, due to the recycling of organic waste, compost can be considered as either an alternative [4] or a fertilizer booster [5]. The utilization of compost as a fertilizer is also in accordance with the EU guidelines for the reduction of non-renewable sources of fertilizer up to 30% (priority axis for Europe 2020) [6].

Thus, the recycling of fertilizer components through composting, as well as the diminished environmental impact due to lowering transportation needs, render this approach as an appealing alternative paving the way towards cyclic economy [7,8]. Last but not least, such approaches offer cost-effective alternatives for poor countries [9].

Onion represents a major source of biowaste, as over the last decade, there has been an increase in trade by at least 25%, reaching 44 million tons and taking the second place in the classification of commercial activity following that of tomatoes [10]. The onion, which belongs to the genus *Allium cepa*, is one of the earliest known fruits. This vegetable, which is widely consumed around the world, is well known for its bioactive content and health-promoting properties. The species are well-known for their sulphated amino acid content, as well as a wide variety of vitamins and minerals. It is also worth noting the presence of secondary metabolites, such as flavonoids, phytosterols, and saponins [11].

Many studies in the literature have used various analytical techniques to determine a wide range of compounds in onion samples. The total phenolic content and antioxidant capacity were determined photometrically [12,13], while specific classes of compounds were determined using liquid chromatographic techniques in conjunction with various detectors, such as UV-Vis [13], diode array (DAD) [14], DAD-MS/MS [15], mass spectrometer (MS) [16], refractive index (HPLC RI Detector) [17], and evaporative light scattering detector (ELSD) [18].

Furthermore, High Resolution Mass Spectrometry (HRMS) techniques, such as (LC-QToF-MS) [15,17,19,20,21], have been used, whereas Nuclear Magnetic Resonance (NMR) [16] has also been employed. Gas chromatography–mass spectrometry (GC-MS) was used for the volatile compounds [19] and fatty acid determination (Golak-Siwulska, Kałużewicz et al., 2018).

*Pleurotus ostreatus* mushroom is a fungus belonging to the *Agaicaceae* family and the *Basidiomycetes* class [22]. *Pleurotus* mushrooms are found in tropical and subtropical forests and are commonly referred to as “oyster mushrooms” [23]. Their widespread distribution makes them the third most extensively produced mushroom around the world [24]. Secondary metabolites, such as phenolic compounds, polyketides, terpenes, and steroids, have been found in mushrooms, all of which are potent antioxidants [25]. *Pleurotus* mushrooms also contain fatty acid esters that exhibit high antioxidant activity and also enhanced antibacterial capacity, preventing the growth of both (+) Gram and (−) Gram bacteria [26].

In the scientific literature, different techniques towards the determination of the total content or specific analyte classes in mushrooms have been documented. Thus, the total phenolic content has been determined photometrically using the Folin–Ciocalteu reagent [27], while selected phenolic compounds were determined using liquid chromatographic techniques in conjunction with DAD [27,28,29,30] or MS detectors [29]. Fungi vitamins were also analyzed using liquid chromatography with a fluorescence detector (FLD) [28]. GC-MS was used to determine volatile chemicals and fatty acids [29] as well as lignans [28]. Electrothermal atomic absorption spectrometry (ETAAS) and inductively coupled plasma mass spectrometry (ICP-MS) were used to measure heavy metals and trace elements [28].

In order to investigate samples with no prior knowledge of their constitution, an arsenal of methodologies has been devised. One of the modes of these methodologies is suspect screening, where a predefined list of potentially existing metabolites in a sample is targeted, usually employing a high-resolution methodology often NMR or HRMS. The list is assembled according to the pre-existing experience of the laboratory, the scientific literature, as well as by the use of heuristic rules, e.g., the metabolic reactions that are involved in the biochemistry of the organism under study. 

Another mode of hypothesis-free experimentation is the non-targeted methodology, where multivariate statistics are employed for the discrimination of the variables that appear up- or down- regulated among the plethora of potential metabolites, which in the case of small molecules is called metabolomics. The combination of these methodologies provides an unbiased holistic view of the chemical potential of unknown samples. The application of hypothesis free screening concerning the composting products has surprisingly caught limited attention [31], although it can be proven as a powerful methodology for the investigation of the composting process. 

Two biofertilizers, using onion and mushroom by-products as raw materials, were produced through composting. In the terms of this study, the possibility that raw material compounds could remain in the corresponding final products through the comprehensive characterization of their common compounds was investigated. The identification of these common compounds was performed using Ultra-High-Pressure Liquid Chromatography-Quadruple Time of Flight Mass Spectrometry (UPLC-QToF-MS) via suspect and non-targeted screening. 

Additionally, for the overall characterization of the aforementioned final products, parameters, such as the pH, conductivity, organic matter, nitrogen content, and elemental analysis, were determined. Such methodologies could contribute towards evaluating sample’s quality control and batch-to-batch reproducibility. Furthermore, the biological role and biochemical content of compost could be meticulously explored. Therefore, these practices could pave the way to the rational optimization of compost production based on the chemical properties.

## 2. Results

Two final products (onion-based and mushroom-based) were derived after the hydrolysis of the compost from the two raw materials. The results derived through the suspect and non-target screening workflows are described in the following sections.

### 2.1. Suspect and Non-Target Screening of the Raw Onion Material and the Onion-Based Final Product

A literature-based suspect list for the onion-based final product was assembled, comprising natural products appearing in onions. A total of eight compounds were identified from this list, both in the raw material and the onion-based final product. Six compounds (myristic acid, palmitic acid, linoleic acid, oleic acid, quercetin, and citric acid) reached identification confidence level 1, whereas two compounds (isorhamnetin-4-glucoside and fumaric acid) were identified at the 2a confidence level.

In total, six compounds were identified through the non-target screening workflow. Τhree compounds (C17-sphinganine, glycerin palmitate, and monostearin) were identified at the confidence level 2a, two compounds (isosakuranetin and x,y-dihydroxybenzaldehyde) reached identification level 3 and one compound reached identification level 5.

The compounds identified by suspect and non-target screening both in the raw material and the final product are tabulated in Table 1 along with their molecular formula, the experimental and predicted retention time, the theoretical and experimental *m*/*z* value of the precursor ion and the ionization mode, as well as the five most abundant MS/MS fragments (if they existed) of the raw material and final product in comparison to the MS/MS fragments of the corresponding spectrum in the spectral library or reference standard.

### 2.2. Suspect Screening and Non-Target Screening of the Raw Mushroom Straw Material and the Mushroom-Based Final Product

Ten compounds were identified from a suspect list compiled using scientific literature for mushrooms and were identified in both the raw material (mushroom straw) and the corresponding final product. Six compounds (myristic acid, palmitic acid, linoleic acid, oleic acid, stearic acid, and leucine) reached the highest identification level (level 1). Furthermore, four compounds (4-hydroxybenzaldehyde, choline, agmatine, and spermidine) were identified at the 2a level of confidence. Moreover, the identification of C17-sphinganine (confidence level 2a) and x,y-dihydroxybenzaldehyde (identification level 3) derived from onions, were components of the final product based on mushroom straw.

Two compounds (9-hydroxy-8,10-dehydrothymol, x,y-dihydroxy-benzaldehyde x = 3, 2, 2, 2 y = 4, 4, 5, 3) were identified through the non-target screening workflow, reaching confidence level 3.

Table 2 illustrates the compounds identified by suspect and non-target screening in both the raw material and the final product. This table also includes their molecular formula, experimental and predicted retention time, theoretical and experimental *m*/*z* value of the precursor ion and ionization mode, and the five most-abundant MS/MS fragments (if they existed) of the raw material and final product in comparison to MS/MS fragments of the corresponding spectrum of the spectral library or reference standard.

### 2.3. Supplementary Chemical Analysis

The results of the physicochemical (conductivity, pH) as well as elemental analysis are summarized in Section A.1 and in Appendix A.

## 3. Discussion

### 3.1. Suspect Screening of the Raw Onion Material and the Onion-Based Final Product

A literature-based suspect list for the onion-based final product was assembled, comprising natural products appearing in onions. [13,14,15,16,17,18,19,21,33,34,35,36]. A total of eight compounds were identified from this list, both in the raw material and the onion-based final product. Six compounds (myristic acid, palmitic acid, linoleic acid, oleic acid, quercetin, and citric acid) reached identification confidence level 1, whereas two compounds (isorhamnetin-4-glucoside and fumaric acid) were identified at the 2a identification confidence level.

Quercetin was used as an example of the methodology of the suspect screening. The theoretical *m*/*z* value of 301.0354 was calculated using Bruker’s Isotope Pattern software, version 4.3 (Bruker Daltonics, Bremen, Germany), and the extracted chromatographic peak for that *m*/*z* value was observed at 7.1 min for both the raw material and the final product. The extracted ion chromatograms of the 4 and 25 eV collision energy were compared concluding that this *m*/*z* value is not a produced ion. In addition, no chromatographic peak for this *m*/*z* value was observed in the blank sample (Figure 1a). Furthermore, the five most abundant ions (higher MW ions than the ion under consideration) for this precursor were investigated to ensure that it is not an in-source fragment. Thus, the *m*/*z* value 301.0354 (±0.005 Da) is not derived from any other ion. Although the chemical structure of the compound is known in the case of suspect screening, the “Smart Formula Manually” software (Bruker Daltonics, Bremen, Germany) was used to evaluate mass accuracy and assign the molecular formula. The neutral molecular formula of C_15_H_10_O_7_ was found for the raw material and the final product was assigned with mass accuracy (2.0 ppm/0.6 mDa, 1.5 ppm/0.4 mDa) and isotope fitting (8.7 mSigma and 40.5 mSigma—a value < 100 mSigma is considered confirmatory of the isotopic fitting), respectively. Furthermore, the compound’s elution time in the raw material and final product (Exp. t_R_ = 7.1 min) was within the 1.8 min confidence margin of the predicted value (t_Rpred_ = 7.0 min) [32]. The Monte Carlo sampling (MCS) method [37] was used to evaluate the difference between the experimental and theoretically predicted time and to eliminate candidates that are false-positive hits (Figure 1b) (can be accessed at the available online tool “Retention Time Indices Platform (RTI platform)” http://rti.chem.uoa.gr/ (accessed on 8 March 2021)). Next, the MS/MS fragments with a signal-to-noise ratio less than 3 were excluded from the MS/MS spectrum list obtained for *m*/*z* 301.0354, and the remaining ones were normalized to be compatible with the MetFrag. This chemical structure corresponds to quercetin and its MS/MS fragments were loaded to MetFrag. All three fragments (107.0143, 151.0036, and 178.9980) were explained by the algorithm. The fragments and their corresponding chemical structures are illustrated in Figure 1c. The MS/MS spectrum of quercetin was compared to the one available in MassBank (MassBank ID: FIO00279). The MS/MS fragments (151.0035 and 178.9986) of the library entry are matched with the experimentally derived fragments (151.0036 and 178.9980) of the suspect compound quercetin in the final product. Following the confirmation of the fragments with the corresponding MS/MS spectra in the spectral library (MS/MS similarity score of 0.961 and 0.870 for the raw material and final product, respectively) and the agreement of the experimental retention time with the predicted one, the confidence level of identification for quercetin assigned to 2a. The common fragments of the final product and the standard spectrum (MassBank ID: FIO00279) are depicted in Figure 1d. Finally, the sample’s MS/MS spectra and elution time were compared to a commercially available standard (t_R_ = 7.1 min). Figure 1e shows the common MS/MS fragments of the product and those of the standard (MS/MS similarity score of 0.961 and 0.922 for the raw material and the final product, respectively). As a result, quercetin was assigned to identification level 1.

For the theoretical *m*/*z* value 227.2017, a chromatographic peak at 13.0 min (raw material and final product) was observed, corresponding to the candidate myristic acid from the suspect list. The neutral molecular formula C_14_H_28_O_2_ was assigned to this mass (mass accuracy of 0.4 ppm/0.1 mDa and 0 ppm/0 mDa) isotopic fit (5.2 and 13.6 mSigma) for the raw material and the final product. The compound was identified using MassBank ID: BS003878 with a level of confidence of 2a. After comparing its MS/MS spectra and retention time with the reference standard (t_R_ = 12.9 min), the identification of myristic acid was confirmed. As a result, myristic acid was assigned the identification level of 1. 

For the theoretical *m*/*z* value 255.2330, a chromatographic peak eluting at 13.8 min for both raw material and final product was detected, corresponding to the candidate palmitic acid. This mass was assigned to a neutral molecular formula of C_16_H_32_O_2_ with mass accuracy of 0.7 ppm/0.2 mDa and 1.0 ppm/0.3 mDa and an isotopic fit of (4.2 mSigma and 7.6 mSigma). Moreover, the predicted retention time (t_Rpred_= 12.9 min) was consistent with the corresponding experimental retention time. The MS/MS fragments matched those in the MoNA spectral Library (MoNA ID: MetaboBASE1104), reaching a confidence level of 2a. The samples’ MS/MS spectra and retention times were then compared to those of the reference standard (t_R_ of standard = 13.8 min), and they were found to be in agreement. As a result, palmitic acid reached a confidence level of 1.

A compound (linoleic acid) from the suspect list was matched for the *m*/*z* value 279.2330, eluting at 13.4 for both the raw material and final product. For this *m*/*z* value, both samples met all of the identification criteria (29.6 and 23.9 mSigma; isotopic fit) and (−0.7 ppm/−0.2 mDa and 0.3 ppm/0.1 mDa; mass accuracy) for the neutral molecular formula of C_18_H_32_O_2_. Furthermore, the predicted retention time corresponded to the experimental results. The MS/MS fragments were evaluated by comparing them to the corresponding reference spectrum in the MassBank library (MassBank ID: RP029511). The MS/MS spectrum and retention time of the reference standard (t_R_ = 13.5 min) were then compared to the samples. Therefore, linoleic acid reached a level of confidence of 1.

The *m*/*z* value 281.2486 is detected in both samples (t_R_ = 13.9 min) and corresponds to oleic acid, which was a candidate in the suspect list. The neutral molecular formula of C_18_H_34_O_2_ was assigned to this mass with mass accuracy of 1.3 ppm/0.4 mDa, isotopic fit of 8.4 and 31.3 mSigma). It is assigned at the level of identification 2a by comparing fragments from the MS/MS spectrum of the reference standard with the attributed in the mass spectral library (MassBank ID: BS003989). The reference standard’s MS/MS spectrum and retention time (t_R_ = 14.0 min) are in close agreement to those of the samples, reaching the level of confidence 1.

A chromatographic peak was observed eluting at 1.1 min (raw material and final product) for the *m*/*z* value 191.0197, which could be attributed to citric acid from the suspect list with assigned neutral molecular formula C_6_H_8_O_7_ (mass accuracy of −7.1 ppm/−1.4 mDa, isotopic fit of 16.9 mSigma) calculated for the feature of the raw material. The identification of citric acid is confirmed after comparing its MS/MS spectra to the corresponding MassBank spectrum with ID: PR100481, as well as to the reference standard. Additionally, the retention time of the reference standard (t_R_ = 1.1 min) was found to be identical to the retention time of the suspect compound in the samples. As a result, citric acid reached level 1.

The theoretical value *m*/*z* 477.1039 corresponds to the suspect compound isorhamnetin-4-glucoside, with elution time of 6.5 min for both compounds (raw material and final product, respectively). This mass was assigned to the neutral molecular formula C_22_H_22_O_12_ (mass accuracy of 1.8 ppm/0.9 mDa and −0.4 ppm/−0.2 mDa; isotopic fit (28.1 mSigma and 7.7 mSigma) for the raw and final product. The predicted retention time (t_Rpred_ = 6.5 min) is consistent with the experimental retention time of the samples. The MS/MS spectrum with MassBank ID: PR040093 was used to identify this compound, yielding a level of confidence of 2a.

The theoretical *m*/*z* value 115.0037, eluting at 1.1 min (raw and final product), corresponds to the candidate fumaric acid from the suspect list. For the raw material, the neutral molecular formula C_4_H_4_O_4_ was assigned (mass accuracy of 0.8 ppm/0.1 mDa, isotopic fit (52.4 mSigma). The predicted retention time (t_Rpred_ = 2.0 min) is in good accordance to the samples’ experimental retention time. After MS/MS evaluation with the corresponding spectrum of MzCloud (No ID: 1274), the compound was identified at the 2a confidence level.

### 3.2. Non–Target Screening of the Onion and Its Corresponding Final Product

In total, six compounds were identified through the non-target screening workflow. Τhree compounds (C17-sphinganine, glucerin palmitate, and monostearin) were identified at the confidence level 2a, two compounds (isosakuranetin and x,y-dihydroxybenzaldehyde) reached identification level 3, and one compound reached identification level 5. The notation followed for the mass spectral data corresponds to the experimental *m*/*z* value _raw material_/*m*/*z* value _final product_.

The *m*/*z* 285.0778/285.0767 chromatographic peak eluted at 9.2 min for both the raw material and final product, with peak scores of 4.41 and 4.66, respectively. The feature was investigated through Smart Formula Manually with the C_16_H_14_O_5_ neutral molecular formula being assigned to this mass (mass accuracy of −3.2 ppm/−0.9 mDa and 0.4 ppm/0.1 mDa; isotope fit of 4.3 mSigma and 16.8 mSigma for the raw material and the final product, respectively). As 195 candidates were retrieved (using the neutral molecular formula and a mass accuracy threshold of 5 ppm) from COCONUT4MetFrag, the retention time prediction tool was used to prioritize the candidates. The dihydroxyflavanone isosakuranetin with predicted retention time of 7.9 min was the most plausible candidate. Six of the eight experimental MS/MS fragments were explained by MetFrag and the MS/MS similarity score for the raw material and the final product with the MassBank Record: BS003552 is 0.731 and 0.727, respectively. However, because the precise position of the hydroxyl moieties on the benzene rings cannot be determined, the highest level of identification that can be reached is 3. For a higher level of identification, the comparison to the corresponding reference standards is required.

One chromatographic peak was observed for *m*/*z* 359.3170/359.3170 at 14.7 for both raw material and final product, with peak scores of 10.7 and 9.58, respectively. The neutral molecular formula C_21_H_42_O_4_ was assigned to this mass (mass accuracy of −4.9 ppm/−1.8 mDa and −4.9 ppm/−1.8 mDa; isotopic fit (27.5 mSigma and 28.7 mSigma for the raw material and the final product, respectively). Employing COCONUT4MetFrag, five candidate structures (neutral molecular formula and 5 ppm mass accuracy) were retrieved with the predicted retention time for monostearin being 13.3 min, and 12 of the 20 experimental fragments (raw material) and 9 of the 17 fragments (final product) were explained by MetFrag. The MS/MS spectra were then compared to the corresponding of the MoNA ID spectrum library: FiehnHILIC001606, and they were found to be in agreement (MS/MS similarity score of 0.732 for the raw material and 0.736 for the final product). As a result, monostearin reached confidence level 2a.

One peak was observed in the chromatograms of the samples (t_R_ 5.5 min and 5.6 min for raw and product, respectively) for 137.0252/137.0258 *m*/*z* values. The neutral mass of C_7_H_6_O_3_ was assigned to it (mass accuracy: −5.5 ppm/−0.7 mDa and −10.3 ppm/−1.4 mDa) with 19 plausible candidates (using a neutral molecular formula, a threshold accuracy of 5 ppm, and the COCONUT4MetFrag Database) through MetFrag. Since all the fragments of the possible candidates (two out of two) were interpreted, their prioritization criteria were the MetFrag score and the retention time prediction. Among these, four isomers scored the highest MetFrag score (1.00), i.e., [3,4-dihydroxybenzaldehyde (t_Rpred_ = 3.7 min), 2,4-dihydroxybenzaldehyde (t_Rpred_ = 4.5 min), 2,5-dihydroxybenzaldehyde (t_Rpred_ = 4.2 min), 2,3-dihydroxybenzaldehyde (t_Rpred_ = 4.1 min). Since the exact position of these hydroxy groups on the benzene ring cannot be defined, the identification level remains at level 3, requiring more evidence.

One chromatographic peak eluting at 10.8 min for both raw material and final product was observed for the *m*/*z* values of 288.2904/288.2903, with peak scores of 5.45 and 9.11, respectively, assigned to the neutral molecular formula C_17_H_37_NO_2_ with mass accuracy of −2.3 ppm/−0.7 mDa and −2.2 ppm/−0.6 mDa, and isotopic fit of 7.6/13.2 mSigma. Using MetFrag one candidate, i.e., C17-sphinganine was retrieved from COCONUT4MetFrag (using the neutral molecular formula and a mass accuracy criterion of 5 ppm). The interpretation of 15 of the 28 experimental fragments for the raw material and 18 of the 34 for the final product was achieved, and the predicted retention time (t_Rpred_ = 11.1 min) was close to the experimental one. The MS/MS spectrum obtained, was compared to the corresponding entry available in the MoNA spectral library (MoNA ID: CCMSLIB00000579284) and found to be in agreement (MS/MS similarity score of 0.822 for the raw material and 0.751 for the final product) reaching the 2a level of identification confidence.

A peak (*m*/*z* value of 331.2850/331.2851) was observed eluting at 14.1 min for both the raw material and its final product, with peak scores of 3.51 and 3.05, respectively. The neutral molecular formula C_19_H_38_O_4_ was assigned to it (−2 ppm/−0.7 mDa, −2.4 ppm/−0.8 mDa, 18.4/23.2 mSigma) and searched using MetFrag in the COCONUT4MetFrag database, deriving seven candidate structures. Among the plausible candidates, 2-palmitoylglycerol, glycerin palmitate, and 2,3-dihydroxypropyl 14-methylpentadecanoate had the highest MetFrag Score and the same number of interpretative MS/MS fragments (11 out of 22 fragments for onion and 11 out of 20 fragments for the final product). The predicted elution time was 12.7, 12.8, and 12.3 min respectively, with glycerin palmitate showing the smallest deviation from the experimental retention value between the candidates. This is reasonable, as the interaction of the glycerin palmitate chain is stronger than these with branches, delaying its elution time [32]. Τhe experimental MS/MS spectrum of glycerin palmitate was compared to the corresponding one in the MoNA spectral library (ID CCMSLIB00000849055) and found to be in accordance (MS/MS similarity score of 0.698 for the raw material and 0.686 for the final product), reaching a confidence level of 2a.

One chromatographic peak was observed for the *m*/*z* 151.0401/151.0402 value at 4.8 min for the raw material and the final product, respectively. However, due to the low peak intensity, its molecular formula could not be matched. COCONUT4MetFrag was used to identify potential candidate structures with this *m*/*z* value and a mass accuracy threshold of 5 ppm. The *m*/*z* value corresponds to 67 candidates, with the proposed neutral molecular formula was C_8_H_8_O_3_. Among the candidate compounds was vanillin, whose predicted retention time (t_Rpred_ = 4.7 min) and reference standard retention time (t_R_ = 4.8 min) were in good agreement with the experimental one of the samples. Thus, this *m*/*z* value can be identified to a confidence level of 5.

### 3.3. Suspect Screening of the Raw Straw Mushroom Material and the Mushroom-Based Final Product

Ten compounds were identified from a suspect list compiled using scientific literature for mushrooms [22,25,27,28,29,30,38,39,40,41,42,43,44,45] and were identified in both the raw material (mushroom straw) and the corresponding final product. The procedure for determining these compounds is briefly outlined below.

For the theoretical value *m*/*z* 227.2017, a chromatographic peak eluting at 13.0 min (both raw and final product) was identified as a suspect compound (myristic acid) in our list. This *m*/*z* value (mass accuracy of 1.0 ppm/0.2 mDa, 0.4 ppm/0.1 mDa) isotopic fit (4.7 mSigma, 17.2 mSigma) was assigned to the neutral molecular formula C_14_H_28_O_2_ for both samples. The compound’s MS/MS spectrum matched that of MassBank ID: BS003878, reaching a 2a level of confidence. The standard’s MS/MS spectrum and elution time (t_R_ = 11.9 min) were compared to those of the samples and found to be in good agreement. Thus, its identification reached the highest level (level 1).

The feature corresponding to the *m*/*z* value 255.2330 (t_R_ raw material = t_R_ final = 13.8 min) is matched to the suspect compound palmitic acid. The neutral molecular formula C_16_H_32_O_2_ was assigned to it, which met all the criteria (mass accuracy = 0 ppm/0 mDa, 1.2 ppm/0.3 mDa, isotopic fit = 13.1 mSigma, 3.2 mSigma) for both samples, and the predicted retention time (t_Rpred_ = 12.9 min) was acceptably close to their corresponding experimental retention time. The candidate’s MS/MS fragmentation matched that of the MoNA spectral library (MoNA ID: MetaboBASE1104) (level 2a). The samples’ MS/MS spectra and retention time were compared to those of the standard (t_R_ of standard = 13.8 min), and they were found to be in agreement. Therefore, palmitic acid reached confidence level 1.

For the *m*/*z* value 279.2330, a chromatographic peak was observed at 13.4 and 13.5 min for the raw material and the final product, respectively, corresponding to linoleic acid, included in the suspect list. The neutral molecular formula C_18_H_32_O_2_ was assigned to this value (mass accuracy of 0.9 ppm/0.2 mDa, 2.4 ppm/0.7 mDa; isotopic fit of 13.1 mSigma, 22.7 mSigma) for the raw material and the final product, and the predicted elution time (t_Rpred_ = 12.9 min) was acceptably close to the experimental ones. The MS/MS fragments were in line with those of the MassBank (MassBank ID: RP029511), reaching a level of confidence of 2a. The reference standard’s MS/MS spectra and elution time (t_R_ = 13.5 min) were identical to those of the samples. These steps confirm the identification of linoleic acid at level 1.

For the theoretical value *m*/*z* 281.2486, a chromatographic peak (t_R_ raw = 13.9 min, t_R_ final = 14.0 min) corresponding to the suspect compound oleic acid was noticed. The neutral molecular formula C_18_H_34_O_2_ was assigned to this value (raw = 0.9 ppm/0.2 mDa, 35.1 mSigma, final = 1.7 ppm/0.5 mDa, 0.2 mSigma). Thus, it was confirmed at the 2a level of identification via the comparison of the experimental to the fragments from the reference standard’s MS/MS spectrum in the mass spectral library (MassBank ID: BS003989). The standard’s MS/MS spectrum and retention time (t_R_ = 14.0 min) were found to be in good agreement with the samples’ respective ones, resulting in the identification of oleic acid at level 1.

For the *m*/*z* value 283.2643, a chromatographic peak (t_R_ raw = 14.4 min and t_R_ final = 14.4 min), corresponding to the suspect compound stearic acid, was detected. The neutral molecular formula C_18_H_36_O_2_ was assigned to this value, fulfilling the criteria of mass accuracy and isotopic fit (raw = 0.2 ppm/0.1 mDa, 3.6 mSigma and final = 1.5 ppm/0.4 mDa, 6.2 mSigma) for both samples. Comparing the fragments to those from the reference spectrum (MassBank ID: BS003930), as well as the agreement of the predicted (t_Rpred_ = 13.4 min) to the experimental retention time, the 2a identification confidence level was reached. Additionally, stearic acid was confirmed by the reference standard and reached the highest level of identification (level 1).

A chromatographic peak was observed at 4.7 min (raw and final product) for the theoretical value *m*/*z* 121.0295, which corresponds to the suspect compound, 4-hydroxybenzaldehyde. The neutral molecular formula C_7_H_6_O_2_ was assigned for the raw material and final product (mass accuracy: 3.8 ppm/0.5 mDa, 1.97 ppm/0.24 mDa, isotopic fit: 2.3 mSigma, 33.1 mSigma, respectively). The predicted elution time (t_Rpred_ = 4.2 min) agrees to the experimental one, and its MS/MS was identified by mzCloud No ID: 234, reaching the 2a confidence level.

The theoretical value *m*/*z* 132.1019 for both samples eluting at 1.7 min belongs to a candidate (leucine) from the suspect list. This mass was assigned the neutral molecular formula C_16_H_13_NO_2_, which met the criteria of mass accuracy (3.3 ppm/0.4 mDa, 4.0 ppm/0.5 mDa) and isotopic fit (9.9 mSigma, 43.1 mSigma) for mushroom straw and its final product, respectively. The predicted retention time (t_Rpred_ = 1.5 min) agreed with the experimental retention time of the samples. Leucine was confirmed at confidence level 2a by comparing its MS/MS spectrum with the one available in the spectral library (mzCloud No ID: 06). The MS/MS spectrum and the retention time of the standard (t_R_ = 1.7 min) agreed with the corresponding ones of the samples, reaching level 1

A chromatographic peak eluting at 1.4 min (raw and final product) corresponds to the candidate compound choline from the suspect list with the theoretical *m*/*z* value of 104.1070. This value was assigned the neutral molecular formula C_5_H_14_NO (mass accuracy of −6.6 ppm/−0.7 mDa, −8.8 ppm/−0.9 mDa for the raw and final products). The predicted retention time (t_Rpred_ = 1.7 min) matched that of the experimental retention time of the samples. Choline was verified by comparing its experimental MS/MS spectrum to the corresponding one in Μass Bank library (ID: PR100405), thereby, gaining the 2a level of confidence.

The theoretical value *m*/*z* 131.1291 eluting at 1.4 min for both samples, corresponds to the candidate compound agmatine. The neutral molecular formula C_5_H_14_N_4_ was assigned to this value (mass accuracy of −6.5 ppm/−0.9 mDa, isotopic profile (4.9 mSigma) for the raw material. The predicted elution time (t_Rpred_ = 0.9 min) was in accordance with the experimental one of the samples. Considering all the above, we concluded that the agmatine was identified by MoNA ID: PT106600 at a level of confidence of 2a.

A chromatographic peak was observed at 1.4 and 1.3 min for the raw material and final product, respectively, which corresponds to the suspect compound spermidine with a theoretical *m*/*z* value of 146.1652. This value was assigned the neutral molecular formula C_7_H_19_N_3_ (mass accuracy of −3.4 ppm/−0.5 mDa for raw and final product, 8.9 ppm/1.3 mDa). The predicted retention time (t_Rpred_ = 0.9 min) matches the samples’ experimental retention time. Spermidine was verified by comparing its experimental MS/MS spectrum to the corresponding one in the MoNA spectral library (ID: MoNA032833), yielding a level of confidence of 2a.

### 3.4. Non-Target Screening of the Raw Straw Mushroom Material and Mushroom-Based Final Product

Two compounds (9-hydroxy-8,10-dehydrothymol, x,y-dihydroxy-benzaldehyde x = 3,2,2,2 y = 4,4,5,3) were identified through the non-target screening workflow, reaching at the confidence level 3. The notation followed for the mass spectral data corresponds to the experimental *m*/*z* value _raw material_/*m*/*z* value _final product_.

One chromatographic peak was observed for the *m*/*z* value 165.0904/165.0904 at 6.5 min for both samples, with peak scores of 5.34 and 5.73, respectively. The neutral molecular formula C_10_H_12_O_2_ was assigned to this value (mass accuracy: −3.80 ppm/−0.63 mDa, −3.59 ppm/−0.59 mDa, isotopic fit: 6.3 mSigma, 6.4 mSigma) for the raw material and the final product. Following that, 112 candidates were retrieved from COCONUT4MetFrag via MetFrag (using the neutral molecular formula and a mass accuracy threshold of 5 ppm). The plausible candidates with the highest MetFragScore (dec-8-en-4,6-diyne-1,3-diol (MetFragScore = 1.00), dec-2-en-4,6-diyne-1,10-diol (MetFragScore = 0.9841), 9-Hydroxy-8,10-Dehydrothymol (MetFragScore = 0.9818) were further investigated. The retention time prediction was used to prioritize the above-mentioned candidates with the following results derived: (dec-8-en-4,6-diyne-1,3-diol (t_Rpred_ = 5.2 min), dec-2-en-4,6-diyne-1,10-diol (t_Rpred_ = 5.4 min), and 9-hydroxy-8,10-dehydrothymol (t_Rpred_ = 6.4 min). The theoretically predicted retention time of 9-hydroxy-8,10-dehydrothymol demonstrated the smallest deviation from the potential candidates. Furthermore, the presence of the diagnostic ion 91.05426 (corresponding to benzyl radical group), as well as the value of 5.5 for the rdb score obtained for the molecular formula indicate the presence of a ring, excluding the other two compounds, which do not contain a ring. For the compound 9-hydroxy-8,10-dehydrothymol, 6 of the 13 MS/MS fragments are explained for the raw material and 8 of the 13 for the final product, reaching to the level of confidence of 3.

For the *m*/*z* values 137.0252 and 137.0258 for the onion raw material and mushroom-based final product, respectively, one chromatographic peak eluted at 5.5 min. The identification process for this *m*/*z* value, which corresponds to x,y-dihydroxy-benzaldehyde (x = 3,2,2,2 y = 4,4,5,3), was discussed in Section 2.3, as this compound was derived to the mushroom-based final product as a result of the contribution of the onion raw material.

### 3.5. Beneficial Role of the Identified Compounds in Plant Health

Plants are constantly exposed to environmental stresses, which can be classified as biotic (e.g., pathogen attack by fungi, bacteria, and herbivores) or abiotic (e.g., drought, radiation, salinity, floods, and extreme temperature conditions) [46]. Two final products (onion- and mushroom-based) were developed to counteract the adverse effects of these conditions and promote plants health. The beneficial role of the specific compounds identified in the final products is briefly discussed.

Free fatty acids’(FFA) ability to kill or inhibit the growth of bacteria render them appealing as antibacterial agents, especially when the use of conventional antibiotics are not preferable or permitted [47]. Additionally, the inhibitory effect of long chain fatty acids against Gram-positive bacteria [48], as well as the involvement of plant-derived C18:0, C18:1, C18:2, and C18:3 (FFA) against a wide range of pathogens and opportunistic microorganisms have been discussed in the scientific literature [49]. Citric acid, belonging to the category of low molecular weight organic acids, is an effective agent for removing significant amounts of heavy metals (Cd, Cu, and Pb) from soil (known as metal phytoextraction) [50].

Moreover, flavonoids may act as attractants or feeding stimulators for some insect species as well as contribute to plant defense [51]. Specifically, quercetin promotes proper plant growth and development by facilitating a variety of physiological processes in plants, including seed germination, pollen growth, and photosynthesis. In addition to the aforementioned beneficial effects, quercetin may also aid in plant resistance to biotic and abiotic stresses due to its high antioxidant capacity [52]. Furthermore, Kurepa et al. recommended incorporating quercetin into growth media due to its beneficial effect on plant stress resistance [53].

Choline could facilitate in plant resistance to osmotic stress (drought and salinity), as well as nutritional value enhancement [54]. Spermidine, which acts as a plant growth regulator, contributes significantly to regulating plants’ stress tolerance [55]. Thus, the presence of the above-mentioned compounds in the final products may improve plant health.

## 4. Materials and Methods

### 4.1. Compost Methodology

In the composting area of the IKORGANIC company production unit, a space was demarcated and cleaned to create composting piles (batch) comprised of chopped raw materials The procedure started with the creation of composting series and the diffusion of oxygen to them via perforated tubes. The temperature was monitored in a daily basis. An increase in temperature indicates the formation of mesophilic microorganisms, which are responsible for the initialization of material fermentation and the denaturation of the compounds thereof. Thermophilic microorganisms formed when the temperature rises above 45 °C. 

After 6–10 days, when the homogenization process was completed, the pile was inverted and placed in a new batch. The heaps were frequently inverted to allow oxygenation, resulting in a constant rise in temperature and the completion of fermentation. The pile was inverted for the second time either at the end of 10–12 days or when the temperature reached 75 °C. The pile’s temperature remained high for 5–10 days before gradually declining. Following the processing of the raw materials and the placement of the piles in the proper temperature-humidity-ventilation conditions, a temperature decrease was observed as cryophilic microorganisms grew. 

When the temperature reached 40–50 °C, the material was removed from the composting square and placed on a waiting maturation area for 2 to 3 weeks, during which it was soaked in water and inverted, the composting process was completed, and the maturation stage for the samples began (Appendix A). The samples were then transitioned to a curing area and covered with cloth for a period of time, allowing their composition to be stabilized. Several parameters, including the type (onion or mushroom) and homogeneity of the mixture, its humidity, how it is cut, and the external weather conditions, may influence the required time for compost competition. Following the maturation process, the products were passed through a sieve with a 2 cm diameter.

### 4.2. Final Products Process

#### 4.2.1. Onion-Based Final Product

The onion compost was hydrolyzed with addition of aqueous glycerin mixture (2:1, *v*/*v*) at 30 °C, yielding a liquid with biostimulant properties.

#### 4.2.2. Mushroom-Based Final Product

The mushroom-based final product comprised of a mixture from the mushroom compost and the onion-based final product in a proportion of 4:1 (*w*/*v*). The above-mentioned was homogenized thoroughly using a mixer. The homogenization procedure lasted 8 days at a constant temperature of 20 °C, and the resulting liquid was the mushroom-based final product.

### 4.3. Reagent and Materials

All standards and reagents used were analytical grade of purity (<95%), unless differently stated explicitly. Methanol (MeOH) (LC–MS grade) was acquired from Merck (Darmstadt, Germany), whereas formic acid 99% and acetic acid were purchased from Fluka (Buchs, Switzerland). Ammonium acetate, ammonium formate, nitric acid (HNO_3_), hydrochloric acid (HCl), sulfuric acid (H_2_SO_4_), phosphoric acid (H_3_PO_4_) and hexane were obtained from Fisher Scientific (Geel, Belgium). The ultrapure water (H_2_O) was provided by a Milli-Q device (Millipore Direct-Q UV, Bedford, MA, USA). Boric acid (H_3_BO_3_) was acquired from Penta (Prague, Czech Republic), while potassium sulfate (K_2_SO_4_), mohrite (Fe(NH_4_)_2_(SO_4_) * 6H_2_O) and ethanol (EtOH) (HPLC grade) were acquired from Acros Organics (Dreieich, Germany). 

Hydrogen peroxide solution 30% (H_2_O_2_) and acetone were acquired from Carlo Erba Reagents (Val de Reuil, France). Potassium dichromate (Κ_2_Cr_2_O_7_) was provided by Panreac Chimica (Barcelona, Spain). Regenerated cellulose syringe filters (RC filters, pore size 0.2 μm, diameter 15 mm) were purchased from Macherey-Nagel (Düren, Germany). Sodium hydroxide (NaOH), fatty acid methylesters, quercetin, citric acid, and leucine were bought by Sigma-Aldrich (Stenheim, Germany). 

Copper (II) sulfate pentahydrate (CuSO4 * 5H_2_O 99%) and vanillin were purchased from Alfa Aesar (Karlsruche, Germany). Stock solutions of the reference standards (1000 mg L^−1^) were prepared in MeOH (LC-MS grade) and stored at −20 °C in ambient glass containers. Working solutions of various concentrations were prepared by appropriate dilutions of the stock solutions with a mixture of MeOH: H_2_O (50:50, *v*/*v*). A multi-element certified reference material (20 analytes including Zn, Cd, V, Fe, Ba, Cu, Ni, B, Pb, Mo, Mn, Cr, and Al) was bought by CPAchem (Zagora, Bulgaria).

### 4.4. Sample Pre-Treatment for HRMS Analysis

Τhe onion and mushroom samples were freeze-dried to achieve compound preconcentration and better preservation. Various parameters affecting the yield of the solid-liquid extraction used as the pretreatment methodology were tested, i.e., the solvent polarity (MeOH, acetone, hexane, and H_2_O in various proportions), the use of extraction assisted sonication, the extraction time and temperature. Τhe description of a sample preparation procedure of the raw materials and their corresponding products for the HRMS analysis can be found in detail in the main text (Section 4.4.2 and Section 4.4.3), while the other sample preparation procedures are detailed in the Section A.2.6.1, Section A.2.6.2, Section A.2.6.3 and Section A.2.6.4.

#### 4.4.1. Freeze-Drying of Raw Materials

The onion samples were homogenized after being chopped into little pieces using a home homogenizer, whilst the mushroom straw was cut into smaller pieces with a pair of scissors. Following, the samples (onion with peel, mushroom straw) were weighed in pre-weighed petri dishes and stored at −80 °C for five hours in order to facilitate freeze-drying. The temperature of the freeze drier was set to −55 °C, and the obtained vacuum was 0.05 mbar. The procedure was completed after 24 h, and the lyophilized samples in the petri dishes were weighed again. The percentage of moisture for the onion and mushroom straw were (90.9 ± 0.1)% and (90.0 ± 0.3)%, respectively.

#### 4.4.2. Sample Preparation of Onion and Its Corresponding Final Product

The same sample preparation was followed for both raw material and its corresponding final product, differentiating only the initial mass of the investigated sample. Therefore, either 0.5 g of a lyophilized onion sample or 2.0 g of the onion-based final product were weighed in 50-mL centrifuge tube. The addition of 10 mL (MeOH: aq 1% formic acid, 80:20 (*v*/*v*)) was followed. Subsequently, they were vortexed for 1 min, followed by shaking employing a horizontal shaker for 30 min. Afterwards, the samples were placed in an ultrasonic bath at 30 °C for one hour and centrifuged at 4000 rpm for ten minutes. The supernatants were then collected, filtered through RC syringe filters, and transferred to 2-mL autosampler glass vials. The extracts were injected into the UPLC-QToF-MS system.

#### 4.4.3. Sample Preparation of Mushroom Raw Material and Its Corresponding Final Product

Considering the higher concentration of bioactive compounds in onions than in mushroom samples reported in the literature [25,28,39,56], a preconcentration step was added to the above-mentioned procedure. As a result, the aforementioned extraction step was performed for these samples, followed by an evaporation step. The supernatants were collected in round-bottom flasks and evaporated at 35 °C using a rotor evaporator. The residues were reconstituted with the addition of 5 mL of EtOH, transferred to glass tubes and evaporated to dryness under a gentle nitrogen stream at 35 °C. Finally, the residues were reconstituted with the addition of 0.5 mL of MeOH: H_2_O (50:50, *v*/*v*), and the extracts were filtered through RC syringe filters, transferred to 2-mL autosampler glass vials, and injected into UPLC-QToF-MS system.

### 4.5. Instrumentation

#### 4.5.1. UPLC-QToF-MS Instrumentation

The raw materials and the corresponding final products were chemically characterized using Ultra High-performance Liquid Chromatography (UHPLC) using an HPG-3400 pump (Dionex Ultimate 3000 RSLC, Thermo Fisher Scientific, Dreieich, Germany) coupled to a time-of-flight mass analyzer (Hybrid Quadrupole time of Flight Matic Bruker Daltonics, Bremen, Germany). An Acclaim RSLC 120 C18 column (2.2 μm, 2.1 × 100 mm, Thermo Fisher Scientific, Dreieich, Germany) was used for the chromatographic analysis, equipped with a pre-column (Van guard Acquity UPLC BEH C18 (1.7 μm, 2.1 × 5 mm, Waters, Ireland). 

The column temperature was kept constant throughout the chromatographic analysis at 30 °C, and the injection volume was set at 5 μL. The analysis was conducted in both positive and negative ionization modes. The mobile phases in the positive ionization mode were composed of: (A) aq. 5 mm ammonium formate: MeOH (90:10 *v*/*v*) acidified with 0.01% formic acid and (B) 5 mM ammonium formate in MeOH acidified with 0.01% formic acid and, in the negative ionization mode, were (a) aq. 10 mM ammonium acetate: MeOH (90:10 *v*/*v*) and (B) 10 mM ammonium acetate in MeOH. The same gradient elution program was used in both ionization modes. 

Initially, the column was equilibrated at 99% (A) at a flow rate of 0.2 mL min^−1^ for 30 min. The elution program was started with 99% (A), with a flow rate 0.2 mL min^−1^ and remained constant for 1 min. Afterward, it was decreased to 61% in 2 min without changing the flow rate. The flow rate was increased to 0.4 mL min^−1^ over the next 11 min, while (A) was reduced to 0.1%. In the next step, the flow rate was increased to 0.48 mL min^−1^, while (A) was kept constant in the ratio of 0.1%. 

Finally, the ratio of the aqueous (A) to organic (B) phases as well as the flow rate were restored to the initial conditions for 3 min before the next injection. Regarding the MS parameters, the values 3500 V for the capillary voltage, 2 bar for the nebulizer gas pressure (N_2_), 8 L min^−1^ for the drying gas, and 200 °C for the capillary temperature were selected. 

The Q-ToF system was calibrated on a daily basis using sodium formate as the calibrant, which was prepared in H_2_O: isopropanol (50:50, *v*/*v*) and injected at the beginning of each run. Both data independent (broad-band Collision Induced Dissociation -bbCID: DIA) and data dependent acquisition (AutoMS, DDA) modes were used for the analysis employing two MS methods. The MS and MS/MS spectra were acquired by applying two different collision energies (4 eV and 25 eV) in the bbCID mode. The five most abundant ions per MS scan were chosen and fragmented with ramp collision energy (collision energy based on *m*/*z* values) in AutoMS mode.

#### 4.5.2. Instrumentation for Additional Analyses

The instrumentation used for the physicochemical (conductivity, pH), the nitrogen and the carbon determination, as well as the elemental analysis, can be found in the Section A.3.

### 4.6. Mass Spectrometry Data Analysis

#### 4.6.1. Identification Confidence

The feature annotation was performed according to the Schymanski et al. scheme, considering the five levels of confidence in identifying a plausible candidate [57]. Starting at confidence level 5, the exact mass of interest was confirmed, although no other evidence for the candidate’s molecular formula exists. The feature’s formula was assigned at the next level of identification confidence (level 4) [58]. At the next level of identification confidence (level 3), a plausible candidate was tentatively identified by evaluating its MS/MS fragments using in silico fragmentation tools (MetFrag [59] or CFM-ID [60]) and its identification was enhanced using prioritization methods, such as retention time prediction [32] and ionization efficiency estimation [61]. 

The existence of a diagnostic ion in a potential candidate may result in identification at level 2b, whereas level 2a is reached when the similarity score between the MS/MS candidate’s spectra and its corresponding spectrum available online spectral libraries is greater than 0.7. Finally, the highest level of identification confidence (level 1) was achieved when the reference spectra and retention time of the candidates were in accordance with those of the reference standard.

#### 4.6.2. Suspect Screening Methodology

The approach of suspect screening is based on ‘prior knowledge’ of the compounds present in the sample considering the corresponding scientific literature. A suspect list was compiled, comprising compounds identified in onions according to the literature using various analytical techniques [13,14,15,16,17,18,19,21,33,34,35,36].The Isotope Pattern (Bruker Daltonics, Bremen, Germany) was used to calculate the exact mass of pseudomolecular ions, [M + H]^+^ for positive ionization (+ESI) and [M − H]^−^ for negative ionization (−ESI). 

A quantitative structure-retention (QSRR) model was also used to predict the plausible candidate’s retention time [32,62]. The initial level for a compound identification in the suspect screening approach is level 3 (level of identification confidence 3), based on the relative confidence scheme reported in the literature [57]. The mass accuracy (expressed in ppm or mDa) and the isotopic profile accuracy (expressed in mSigma) [63,64] were evaluated using the ‘Smart Formula’ software (which is part of Data Analysis, Bruker Daltonics, Bremen, Germany). 

The molecular formula was assigned to the mass value (*m*/*z*) by mass tolerance of 2 mDa and isotopic tolerance of 100 mSigma. Τhe MetFrag in silico fragmentation software [65] was used to assist the identification of the compounds by interpreting MS/MS fragments of this compound [65]. The MS/MS spectra of suspect compounds were searched in the literature and spectral libraries (MoNA [66], MassBank [67], GNPS [68], and mzCloud [69] (confidence level 2). 

A QSRR, model was also used to predict the candidate molecule’s retention time [32,62] developed in-house based on the proposed structure, physicochemical properties, and chromatographic system (analytical column, elution program and mobile phases). The elution time was predicted for all compounds that reached the 2a, 2b, and 3 confidence levels, and the compounds were classified into a different box (an example is illustrated in Figure 1b) based on the criteria listed in the literature [32]. Finally, the MS/MS spectra of the suspect compounds were compared to those of the authentic reference standards (confidence level 1), if available in the laboratory. Since the analytical data available to identify the targeted *m*/*z* values varied, the identification reached different levels of confidence.

#### 4.6.3. Non-Target Screening Methodology

##### Peak Picking and Initial Screening

Non-targeted screening was performed to identify common compounds found in raw materials and their corresponding final products using the open-source MS Dial software (version 4.60) [70]. Initially, the raw data was calibrated in Data Analysis software (Bruker Daltonics, Bremen, Germany) and converted by the ABF converter [71] to be compatible with the peak picking software. The ions [M + H]^+^, [M + NH_4_]^+^, [M + Na]^+^, [M + K]^+^, [M + CH_3_OH + H]+, and [M + NH_3_ + H]^+^ were tested for the positive and [M − H]^−^, [M + H_2_O − H]^−^, [M + Cl]^−^, [M + HCOOH − H]^−^, and [2M − H]^−^ and for the negative ionization mode. The peak lists were defined as: ‘All public MS/MS positive’, which contained 13.303 unique compounds for the positive and ‘All public MS/MS negative’, which contained 12.879 unique compounds for the negative ionization mode [72].

##### Peak Annotation

Peak annotation was performed to assign the detected features to plausible candidates. The workflow is described below. Peak scores, defined as the ratio of the candidate peaks area to its corresponding peak intensity, have been calculated for the *m*/*z* values of interest. A value > 4 is considered acceptable [63]. Following that, the assigned formula’s mass accuracy and isotopic profile were evaluated using the Smart Formula Manually software (Bruker Daltonics, Bremen, Germany) to achieve a level of confidence 4. The candidates (using the molecular formula of the neutral species showing ≥ 5 ppm mass accuracy) were retrieved from the COCONUT4MetFrag database and proceeded to MetFrag for obtaining the in silico MS/MS predicted spectra. 

It is worth noting that COCONUT (the Collection of Open Natural Products) is an open database of 406.744 natural products [73]. The candidates were then prioritized using an in-house retention time prediction tool (confidence level 3) [32,63]. The MS/MS spectra of the suspect compounds were then searched against the literature and mass spectra libraries (MoΝA [66], MassBank [67], GNPS [68], and mzCloud [69] confidence level 2). In cases where neither the MS/MS spectrum could be retrieved from the online libraries nor the standard compound was available, in silico fragmentation was performed, and the most abundant explained fragments were evaluated. 

The MS/MS similarity score for each compound should be greater than 0.7 in order to be annotated. The elution time was predicted for all compounds that reached the 2a, 2b, and 3 confidence levels, and the compounds were classified into a different box (an example is illustrated in Figure 1b) based on the criteria listed in the literature [32]. Finally, where reference standards were available in the laboratory, MS/MS spectra were compared to those of suspect compounds (confidence level 1), with similarity score > 0.7.

## 5. Conclusions

The composting process was applied for the production of two biofertilizers using onions and mushroom by-products as raw materials. These two raw materials and their corresponding final products (onion-based and mushroom-based) were characterized following suspect and non-target methodologies through UPLC-QToF-MS, and the common compounds were elucidated between the investigated samples. In total, 14 compounds were identified in both the onion raw material and the onion-based final product, while 12 compounds were found in both mushroom and the mushroom-based final product. In the latest mentioned product, the contribution of the onion was also taken into consideration as it was a product component. 

In both cases, the identified compounds, belonging to various categories, such as fatty acids, organic acids, flavonoids, and amino acids, enhancing their beneficial effects to plant health. The fatty acids appeared to be the most resilient compounds during the compost process as the majority of the identified compounds belong to this class, while antioxidants appeared to decrease as expected. Additionally, parameters, such as the pH, conductivity, organic matter, nitrogen content, and elemental analysis, were determined for the overall characterization of the aforementioned final products. The results indicate that the presence of the identified compounds could ameliorate plant health.

## Figures and Tables

**Figure 1 molecules-27-03498-f001:**
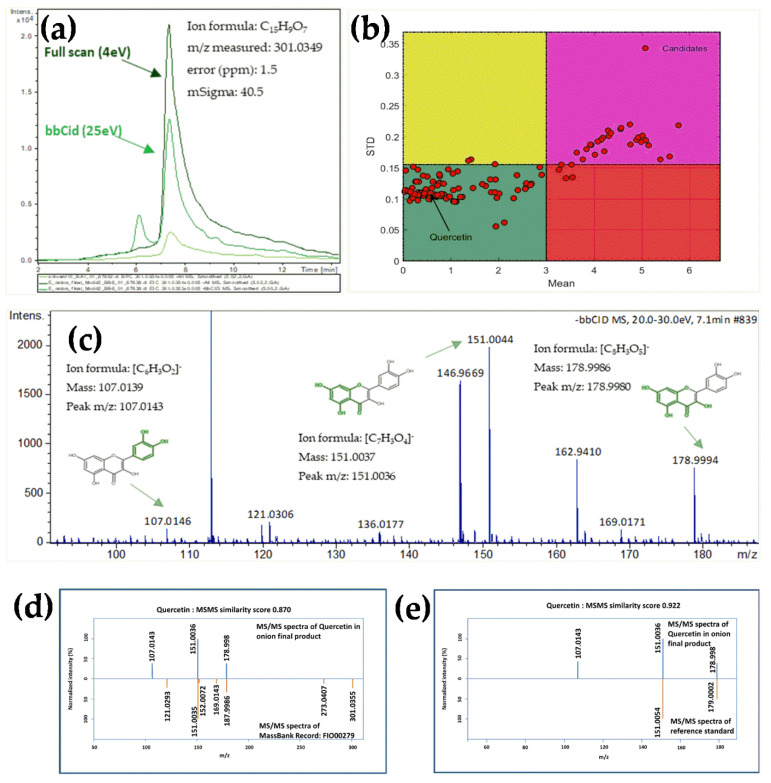
Identification of suspect compound of quercetin: (**a**) fullscan MS and bbCID chromatogram for *m*/*z* value 301.0354 (±5 mDa). (**b**) MCS plot. (**c**) MS/MS spectrum with in silico fragmentation with the corresponding structures and fragments. Comparison of MS/MS spectra of the final product with (**d**) MS/MS spectrum from the spectrum library and (**e**) MS/MS from the reference standard.

**Table 1 molecules-27-03498-t001:** Compounds identified through suspect and non-target screening in the raw material (onion) and onion-based final product.

Compound Name	Chemical Formula	Exp. t_R_ (min) Raw/Final (Reference Standard) ^a^	Pred. t_R_ (min)	Application Domain ^b^	Exp. *m*/*z* Values ^c^ Raw/Final	Theor. *m*/*z*	ESI Mode	Screening	MS/MS Explained Fragments ^d^ (Onions)	MS/MS Explained Fragments ^d^ (Final Product)	Reference MS/MS Spectra ^e^	Level of Identification/ Database Reference ^f^
Myristic acid	C_14_H_28_O_2_	13.0/13.0 (12.9) ^a^	11.9	Region B	227.2016/ 227.2016	227.2017	−ESI	suspect	209.1923	209.1928	209.2007	1
									227.202	227.2015	227.2027	
									228.2054	228.2046	228.2038	
									229.1974	229.2081	229.2030	
Palmitic acid	C_16_H_32_O_2_	13.8/13.8 (13.8) ^a^	12.9	Region A	255.2328/ 255.2327	255.2330	−ESI	suspect	255.2328	255.2326	255.2327	1
									256.2366	256.2361	256.2364	
									257.2409	257.2413	257.2395	
Linoleic acid	C_18_H_32_O_2_	13.4/13.5	12.9	Region A	279.2331/	279.2330	−ESI	suspect	279.2323	279.2335	279.2328	1
		(13.5) ^a^			279.2329				280.237	280.2374	280.2333	
Oleic acid	C_18_H_34_O_2_	13.9/14.0 (14.0) ^a^	13.2	Region A	281.2482/ 281.2485	281.2486	−ESI	suspect	281.2493	281.2487	281.2468 282.2508	1
									282.2528	282.2526		
									283.2653	283.2634		
Quercetin	C_15_H_10_O_7_	7.1/7.1 (7.1) ^a^	7.0	Region A	301.0348/ 301.0349	301.0354	−ESI	suspect	65.0035		65.0054	1
									121.0309		121.0299	
									151.0048	151.0036	151.0054	
									178.9996	178.9980	179.0002	
									301.0358		301.0385	
Citric acid	C_6_H_8_O_7_	1.1/1.1 (1.1) ^a^	1.3	Region A	191.0197/ 191.0173	191.0197	−ESI	suspect	78.9592		78.9593	1
									85.0294	85.0299	85.0297	
									87.0087	87.0088	87.0089	
									111.0049	111.0085	111.0088	
									158.9254		158.9255	
Isorhamnetin-4-glucoside	C_22_H_22_O_12_	6.5/6.5	6.5	Region A	477.1030/ 477.1040	477.1039	−ESI	suspect	151.0096	151.0040	151.0033	2a MassBank ID: PR040093
									243.1064	243.1009	243.0291	
									300.0276	300.0289	300.0262	
									314.0432	314.0390	314.0425	
									477.1034	477.1032	477.1033	
Fumaric acid	C_4_H_4_O_4_	1.1/1.1	2	Region A	115.0036/ 115.0035	115.0037	−ESI	suspect	71.0140	71.0132	71.01385	2a MzCloud no 1274
									79.9573	79.9574		
									88.9880	88.9875		
									96.9598	96.9598		
									115.0036	115.0069	115.0037	
x,y-Dihydroxybenzaldehyde x = 3, 2, 2, 2 y = 4, 4, 5, 3	C_7_H_6_O_3_	5.5/5.6	3.7	Region B	137.0252/ 137.0258	137.0244	−ESI	Non-target	92.0271 108.022 137.0252	92.0255 108.0227 137.026		3
			4.5	Region A								
			4.2	Region B								
			4.1	Region B								
Isosakuranetin	C_16_H_14_O_5_	9.2/9.2	7.9	Region B	285.0778/ 285.0767	285.0768	−ESI	Non-target	151.0042	151.0040	151.0043	3 MassBank ID: BS003552
									164.0118	164.0125	164.0124	
									196.0012	196.0020	196.0020	
									285.0762	285.0765	285.0776	
Monostearin	C_21_H_42_O_4_	14.7/14.7	13.3	Region B	359.3170/ 359.3170	359.3156	+ESI	Non-target	57.0695	57.0694	57.0697	2a MoNA ID: FiehnHILIC001606
									71.0851	71.0854	71.0819	
									83.0855	83.8610	83.0823	
									95.0866	95.0860	95.0827	
									341.3058	341.3055	341.2972	
C17-sphinganine	C_17_H_37_NO_2_	10.8/10.8	11.1	Region A	288.2904/ 288.2903	288.2897	+ESI	Non-target	88.0756	88.0755	88.0755	2a MoNA ID: CCMSLIB00000579284
									106.0871	106.0870	106.0863	
									270.2790	270.2790	270.2787	
									288.2898	288.2896	288.2902	
									289.2937	289.2931	289.2934	
Glucerin palmitate	C_19_H_38_O_4_	14.1/14.1	12.8	Region B	331.2850 331.2851	331.2843	+ESI	Non-target	71.0851	71.0853	71.0851	2a MoNA ID CCMSLIB00000849055
									85.1010	85.1010	85.1017	
									95.0859	95.0866	95.0851	
									239.2361	239.2370	239.2373	
									313.2732	313.2735	313.2743	
Unknown	C_8_H_8_O_3_	4.8/4.8			151.0401		−ESI	Non-target				5
					151.0402							

^a^ retention time of the reference standard, ^b^ Monte Carlo sampling technique (see reference [32]), ^c^ experimental *m*/*z* value with error ± 0.005 Da; [M + H]^+^ for +ESI and [M − H]^−^ for −ESI, ^d^ top-five most intense explained peaks (if they existed), ^e^ MS/MS fragments of the spectra reference standard or mass spectral library, ^f^ for the level of identification 2a the database entry is referred.

**Table 2 molecules-27-03498-t002:** Compounds identified through suspect and non-target screening in the raw materials and the mushroom-based final product.

Compound Name	Chemical Formula	Exp. t_R_ (min) Raw/Final (Reference Standard) ^a^	Pred. t_R_ (min)	Application Domain ^b^	Exp. *m*/*z* Values ^c^ Raw/Final	Theor. *m*/*z*	ESI Mode	Screening	MS/MS Explained Fragments ^d^ (Onions)	MS/MS Explained Fragments ^d^ (Mushrooms)	MS/MS Explained Fragments ^d^ (Final Product)	Reference MS/MS Spectra ^e^	Level of Identification/ Database Reference ^f^
Myristic acid	C_14_H_28_O_2_	13.0/13.0 (12.9) ^a^	11.9	Region B	227.2014/ 227.2016	227.2017	−ESI	Suspect	227.202	227.2016	227.2019	227.2027	1
									228.2054	228.2047	228.2049	228.2038	
									229.1974	229.2052	229.2058	229.2030	
Palmitic acid	C_16_H_32_O_2_	13.8/13.8 (13.8) ^a^	12.9	Region A	255.2330/ 255.2326	255.2330	−ESI	Suspect	255.2328	255.2329	255.2329	255.2327	1
									255.7261	255.7258	255.7258	255.7571	
									256.2366	256.2362	256.2369	256.2364	
									257.2409			257.2395	
Linoleic acid	C_18_H_32_O_2_	13.4/13.4	12.9	Region A	279.2327/	279.2330	−ESI	Suspect	279.2335	279.2332	279.2329	279.2328	1
		(13.5) ^a^			279.2323								
Oleic acid	C_18_H_34_O_2_	13.9/13.9 (14.0) ^a^	13.2	Region A	281.2484/ 281.2481	281.2486	−ESI	Suspect	281.2493	281.2481	281.2488 282.2523	281.2468 282.2508	1
									282.2528	282.2514			
									283.2653	283.2608			
Stearic acid	C_18_H_36_O_2_	14.4/14.4	13.4	Region A	283.2642/	283.2643	−ESI	Suspect	283.2637	283.2639	283.2633	283.2643	1
		(14.5) ^a^			283.2638				284.2674	284.2676	284.2672	284.2677	
4-hydroxy- benzaldehyde	C_7_H_6_O_2_	4.7/4.7	4.2	Region A	121.0030/ 121.0295	121.0295	−ESI	Suspect	nd	92.0269	92.0277	92.0268	2a MzCloud No: 234
										93.0342		93.0346	
										95.0139	95.0145	95.0139	
										108.0215	108.0227	108.0217	
										121.0300		121.0295	
Leucine	C_6_H_13_NO_2_	1.7/1.7 (1.7) ^a^	1.5	Region A	132.1015/ 132.1014	132.1019	+ESI	Suspect	86.0962	86.0962	86.0965	86.0962	1
									87.0437	87.0434	87.0440	87.0434	
									89.0594	89.0592	89.0593	89.0592	
									93.0450	93.0446	93.0447	93.0446	
									132.1018	132.1012	132.1016	132.1012	
Choline	C_5_H_14_NO	1.4/1.4	1.7	Region A	104.1077/ 104.1086	104.1070 *	+ESI	Suspect	58.0648	58.0649	58.0648	58.0658	2a MassBank ID PR100405
									60.0804	60.0805	60.0806	60.0815	
									104.1077	104.1073	104.1082	104.1075	
Agmatine	C_5_H_14_N_4_	1.4/1.4	0.9	Region A	131.1290	131.1291	+ESI	Suspect	n.d.	60.0555	60.0556	60.0570	2a MoNA ID PT106600
										72.0805	72.0804	72.0814	
										114.1023	114.1052	114.1031	
										131.1288		131.1296	
Spermidine	C_7_H_19_N_3_	1.4/1.3	0.9	Region A	146.1647/ 146.1665	146.1652	+ESI	Suspect	n.d.	72.0807	72.081	72.0804	2a MoNA ID MoNA032833
										129.1382	129.1385	129.1387	
										146.1647	146.1665	146.1647	
9-Hydroxy-8,10-Dehydrothymol	C_10_H_12_O_2_	6.5/6.5	6.4	Region A	165.0904/ 165.0904	165.0910	+ESI	Non-target	n.d.	91.0541	91.0547		3
										119.0853	119.0852		
										128.0612	128.0611		
										129.0691	129.069		
										147.0795	147.0796		
x,y-Dihydroxy- benzaldehyde x = 3, 2, 2, 2 y = 4, 4, 5, 3	C_7_H_6_O_3_	5.5/5.5	3.7	Region B	137.0252/ 137.0258	137.0244	−ESI	Non-target	92.0271 108.022 137.0252	n.d.	92.0271 108.0217 137.0251		3
			4.8	Region A									
			4.2	Region B									
			4.1	Region B									

^a^ retention time of the reference standard, ^b^ Monte Carlo sampling technique (see reference [32]), ^c^ experimental *m*/*z* value with error ± 0.005 Da; [M + H]^+^ for +ESI and [M − H]^−^ for −ESI (* with the exception of choline, which *m*/*z* value corresponds to [M]^+^), ^d^ top-five most intense explained peaks (if they existed), ^e^ MS/MS fragments of the spectra reference standard or mass spectral library, ^f^ for the level of identification 2a the database entry is referred.

## Data Availability

Data are available upon request.

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
