# Peer review of "From By-Products to Fertilizer: Chemical Characterization Using UPLC-QToF-MS via Suspect and Non-Target Screening Strategies"

_molecules, 2022, doi:10.3390/molecules27113498_

Round 1

Reviewer 1 Report

The authors proposed a new method of analysis of onion-based and mushroom-based materials being components of fertilizer. All the results obtained were described and thoroughly discussed. The content of the work, however, is very extensive, which may discourage the reader. Of course, I am conscious of the amount of research done and the results analyzed.... but maybe something could be moved to the Supplementary Part, or the text could be shortened a bit? However, the decision in this matter is left to the editors of the magazine. The work is very valuable, and because the methodology is presented in such detail, it can be used by other researchers for its application to another class of compounds.
I recommend the work for publishing.

Reviewer 2 Report

The manuscript writing and the discussion of results against publication of article in its current form. I think that the paper is publishable in the Molecules after some major revisions. The comments are as follows:

  • Title and author affiliation:

The type of the paper should be given.

“Affiliation 1;” should be deleted.

  • Introduction

The introduction gave a satisfactory literature survey on the similar topic and it outlined the proposed method well.

The authors should give prospects for the application of the results of this study.

Line67: The full spelling of “HRMS” should be given.

  • Results

Table 1 & Table 2: Please pay attention the formats of these two tables. Are the vertical lines required?

  • Discussion

The discussion section was not well written. The authors just made some descriptions of the compounds identified, the effects of these compounds on plant growth should be discussed in this section. Even there was no cited references except Line 178.

Line190: Figure 1 should not appear in the discussion section.

Figure 1(d) and (e): Not clear enough for publication. Same in Figure S3.

  • Materials and Methods

The section was well written.

  • References:

References should be checked again according to the journal format.

Line 758 “”” should be deleted.

Reviewer 3 Report

Reviewer report on manuscript Molecules-1733135

The submitted approach provides data regarding to chemical characterization of the compounds that pass from the raw materials to the final products and especially on onion- and mushroom-based products.

The manuscript is well-written, referenced and figured. The novelty of the proposed approach is sufficient according to the Journals’ standards. I have no comments on the experimental part and I therefore recommend acceptance after minor revision.

Comments

  • The references’ style in the main text should be corrected according to the Journals’ guidelines.
  • Which software was used for the Monte-Carlo sampling? This information should be added in the text.
  • Line 684: correct to “[M+H2O-H]-“.

Reviewer 4 Report

Interesting article. I suggest shortening the introduction. also standardize the style of the bibliography. In line 805 there is a typo. In some citations the first name is extended, in others abbreviated.

Round 2

Reviewer 2 Report

The revised manuscript can be accepted.